# Identification of limb-specific *Lmx1b* auto-regulatory modules with Nail-patella syndrome pathogenicity

Endika Haro [1,2], Florence Petit [3,4], Charmaine U. Pira[1], Conor D. Spady[1], Sara Lucas-Toca[2], Lauren I. Yorozuya[1], Austin L. Gray [1], Fabienne Escande[4,5], Anne-Sophie Jourdain[4,5], Andy Nguyen [1], Florence Fellmann [6], Jean-Marc Good [6], Christine Francannet[7], Sylvie Manouvrier-Hanu [3,4], Marian A. Ros [2✉] & Kerby C. Oberg [1✉]

*LMX1B* haploinsufficiency causes Nail-patella syndrome (NPS; MIM 161200), characterized by nail dysplasia, absent/hypoplastic patellae, chronic kidney disease, and glaucoma. Accordingly in mice, *Lmx1b* has been shown to play crucial roles in the development of the limb, kidney and eye. Although one functional allele of *Lmx1b* appears adequate for development, *Lmx1b* null mice display ventral-ventral distal limbs with abnormal kidney, eye and cerebellar development, more disruptive, but fully concordant with NPS. In *Lmx1b* functional knockouts (KOs), *Lmx1b* transcription in the limb is decreased nearly 6-fold, indicating autoregulation. Herein, we report on two conserved <u>*Lmx1b*</u>-<u>a</u>ssociated *cis*-<u>r</u>egulatory <u>m</u>odules (*LARM1* and *LARM2*) that are bound by Lmx1b, amplify *Lmx1b* expression with unique spatial modularity in the limb, and are necessary for Lmx1b-mediated limb dorsalization. These enhancers, being conserved across vertebrates (including coelacanth, but not other fish species), and required for normal locomotion, provide a unique opportunity to study the role of dorsalization in the fin to limb transition. We also report on two NPS patient families with normal *LMX1B* coding sequence, but with loss-of-function variations in the *LARM1/2* region, stressing the role of regulatory modules in disease pathogenesis.

[1] Department of Pathology and Human Anatomy, Loma Linda University School of Medicine, Loma Linda, CA, USA. [2] Instituto de Biomedicina y Biotecnología de Cantabria, CSIC–SODERCAN-Universidad de Cantabria, Santander, Spain. [3] Clinique de Génétique, CHU Lille, F-59000 Lille, France. [4] EA7364 RADEME, Université de Lille, F-59000 Lille, France. [5] Laboratoire de Biochimie et Biologie Moléculaire, CHU Lille, F-59000 Lille, France. [6] Service de Médecine Génétique, Centre Hospitalier Universitaire Vaudois, Lausanne, Switzerland. [7] Service de génétique médicale, CHU Estaing, Clermont-Ferrand, France. ✉email: marian.ros@unican.es; koberg@llu.edu

The LIM homeodomain transcription factor Lmx1b is responsible for limb dorsalization. In the limb, Lmx1b is induced by Wnt7a from the dorsal ectoderm, and its expression is restricted to the dorsal mesoderm[1,2]. Loss of Lmx1b function in mice results in loss of dorsal autopod (hand/foot) and zeugopod (forearm/leg) patterning; the autopods have a symmetrical ventral-ventral phenotype with dorsal footpads, loss of dorsal hair follicles, absence of nails, and a symmetrical ventral-ventral pattern of muscles, tendons and ligaments. Besides the limb, mice lacking functional Lmx1b exhibit abnormal eye, cerebellar, and kidney development which accounts for the perinatal lethality[3]. In contrast to mice, single allele variations in humans that disrupt LMX1B function cause Nail-patella syndrome (NPS; MIM 161200)[4,5]. This autosomal dominant condition is characterized by nail dysplasia, absent or hypoplastic patellae, bone fragility, premature osteoarthritis, chronic kidney disease, and ocular anomalies. Evaluation of the variety of human *LMX1B* mutations indicate that NPS is due to haploinsufficiency[5,6]. Thus, a sub-threshold level of LMX1B is responsible for the syndromic features and incomplete limb dorsalization. In the murine model, homozygous KO mice exhibit a more dramatic phenotype than the human condition, with ventral-ventral distal limbs suggesting a threshold-mediated effect on limb dorsalization.

In the absence of Lmx1b function, transcription of the *Lmx1b* mRNA is decreased nearly sixfold in developing (e12.5) mouse limbs suggesting that one function of Lmx1b is the auto-amplification of its own expression (Supplementary Table 1)[7]. Lmx1b-targeted chromatin immunoprecipitation combined with high-throughput sequencing (ChIP-seq) during limb dorsalization (e12.5) identified two highly conserved Lmx1b-bound *cis*-regulatory modules (CRMs) 60 kb upstream of the *Lmx1b* gene[8]. CRMs are DNA sequences enriched in transcription factor binding sites that regulate associated genes in a time- and tissue-specific manner. Lmx1b-binding to CRMs upstream of its own coding sequence provides a mechanism by which Lmx1b could auto-amplify its own expression.

In this study, we have characterized and functionally validated these two *Lmx1b* associated regulatory modules that we term *LARM1* and *LARM2*. We show that they are highly conserved across vertebrates including coelacanth, but not other fish species. The activity of these two *LARM* sequences overlaps the expression pattern of *Lmx1b* in the dorsal limb mesoderm when assessed either together or individually using both chick and mouse enhancer assays. Removal of the *LARM* region with CRISPR-Cas9 results in a limb phenotype similar to that of animals lacking functional Lmx1b, with marked reduction in *Lmx1b* expression and loss of limb dorsoventral asymmetry, but without any other Lmx1b-related organ system affected. These data establish *LARM1* and *LARM2* as limb-specific *Lmx1b* enhancers necessary for amplifying the level of *Lmx1b* expression in the limbs. Interestingly, about 10% of patients with the NPS phenotype lack a variation in the *LMX1B* coding sequence[9]. We investigated two NPS patient families that lack coding sequence changes but instead have *LARM* variations that disrupt human *LARM* activity, highlighting the important role of *cis*-regulatory modules in development and disease pathogenesis.

## Results

### *Lmx1b*-associated regulatory modules are activated by LMX1B.

We recently identified two Lmx1b-bound CRMs, *LARM1* and *LARM2*, 60 and 66 kb upstream of the mouse *Lmx1b* gene, respectively (Fig. 1a)[8], that are associated with active chromatin marks (p300[10], H3K27ac[11], H3K4me2[12], RNAPOL2, and Med12[13]) during limb development (Fig. 1b). Interestingly, *LARM*2, the more distant module, also associates with inhibitory

marks (H3K27me3)[12]. For a given cell, histone acetylation (H3K27ac) and methylation (H3K27me3) are mutually exclusive; their coexistence indicates tissue heterogeneity. This pattern is consistent with *LARM2* being only accessible and active in the dorsal limb compartment[14].

Using GFP reporter constructs in chicken electroporation bioassays[15,16], we found that both *LARM1* and *LARM*2 demonstrate enhancer activity within the dorsal limb mesoderm coincident with the *Lmx1b* expression domain (Fig. 2a, f). Conservation analysis using multi-species alignment[17] (Fig. 1b) subdivided *LARM1* into two conserved regions, a 5′ element containing one conserved potential Lmx1b binding site and a 3′ element with three conserved binding sites (based on the reported TMATWA binding motif)[8] (Fig. 2a). Surprisingly, the isolated 5′ *LARM1* element did not show reporter activity, whereas the isolated 3′ *LARM1* element showed strong activity in the limb mesoderm but with no dorsal bias (Fig. 2c). Interestingly, the restriction of *LARM1* activity to the dorsal mesoderm requires the Lmx1b binding site within the 5′ element as site-directed mutagenesis expanded enhancer activity into the ventral mesoderm (Fig. 2d). To ensure that a new permissive binding site had not been added, we generated two additional mutants of this Lmx1b binding site in the 5′ element of *LARM1* and both also showed expanded enhancer activity into the ventral mesoderm (Supplementary Fig. 1). In contrast, mutation of any of the three predicted Lmx1b binding sites in the 3′ element resulted in markedly reduced yet still dorsally restricted *LARM1* activity (Fig. 2d–e). These findings are counterintuitive since Lmx1b is only expressed in the dorsal limb mesoderm. A possible interpretation is that the putative Lmx1b binding site within the 5′ *LARM1* element (TTATTA) can bind other transcription factors or corepressors that silence the 3′ *LARM1* enhancer activity or promote chromatin conformation that limits enhancer-promoter interaction[18]. In the dorsal mesoderm, Lmx1b would compete for this binding site, counteracting the silencer function of the 5′ *LARM1* element, and drive enhancer activity. In support of this view, human LMX1B activates this enhancer when expressed together ectopically in the ventral limb bud mesoderm (Fig. 2j). Collectively, our data indicate that *LARM1* is composed of a 3′ enhancer (*LARM1*e) and a 5′ silencer (*LARM1*s) that blocks ventral activity, thereby restricting its function to the dorsal limb.

In contrast, *LARM*2 is composed of a single conserved element containing two highly conserved Lmx1b binding sites (Fig. 2g). The dorsally restricted activity of *LARM*2 is abolished by disruption of either binding site (Fig. 2h, i) suggesting that *LARM*2 is a positive *Lmx1b* enhancer whose activity depends on Lmx1b. Furthermore, the observation that *LARM*2 reporter activity is activated in the ventral limb mesoderm by ectopically expressing human LMX1B fully corroborates this conclusion (Fig. 2j). These results, together with the pattern of chromatin marks in the *LARM2* enhancer (Fig. 1b), suggest that Lmx1b plays a role in chromatin/*LARM2* activation within the dorsal compartment.

Thus, our results show that both *LARM1* and *LARM*2 are bound by Lmx1b[8], display dorsal restricted activity in limb buds, require Lmx1b binding sites for activity, and are activated by human LMX1B when co-expressed in the ventral mesoderm. This, together with published capture C experiments[19] showing that the *LARM* region physically interacts with the *Lmx1b* promoter (Supplementary Fig. 2), supports the concept that *LARM1* and *LARM*2 are bona fide *Lmx1b* autoregulatory enhancers.

We note that the *Lmx1b* locus includes a long non-coding RNA (C130021I20) (https://doi.org/10.1371/journal.pone.0028358; http://www.biomedcentral.com/1471-213X/11/47) that is transcribed

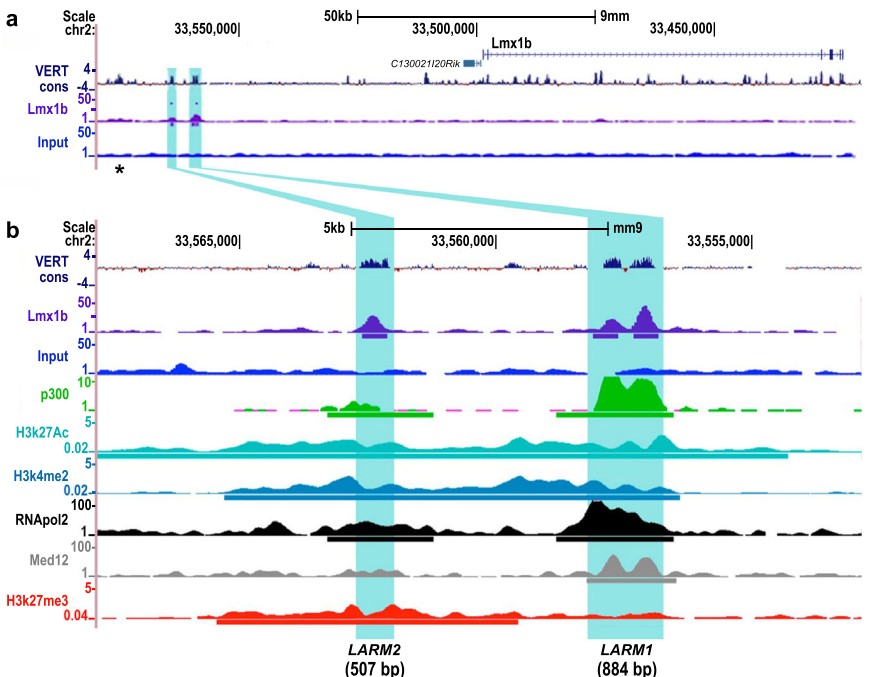

**Fig. 1 LARM1 and LARM2 are conserved, bound by Lmx1b, and associated with active chromatin marks. a** UCSC genome screenshot displaying the *Lmx1b* locus and the associated *cis*-regulatory modules *LARM1* and *LARM2* (highlighted in blue) showing vertebrate conservation (VERT cons), Lmx1b binding (Lmx1b-targeted ChIP-seq)[8] and input[8]. **b** Magnification of the putative enhancer region displaying the overlap with active enhancer-associated regulatory marks present in limb buds. From top to bottom: vertebrate conservation (VERT cons), ChIP-seq tracks for Lmx1b[8], control Input limb DNA[8], p300[10], histone 3 acetylation at lysine 27 (H3K27Ac)[11], histone 3 dimethylation at lysine 4 (H3K4me2)[12], RNA polymerase II[13], Med 12[13], and histone 3 trimethylation at lysine 27 (H3K7me3)[12]. Note that *LARM1* consists of two conserved peaks, both of which are recognized by Lmx1b-targeted ChIP-seq[8]. Upstream of *LARM2*, another conserved region is present (asterisk). This potential *cis*-regulatory module is also associated with the 9430024E24Rik gene[21], but does not appear to be bound by Lmx1b, as shown in the Lmx1b-targeted ChIP-seq track (see also Supplementary Fig. 2).

from the opposite strand of *Lmx1b* using the same bidirectional promoter[20,21]. This lncRNA transcript shows the same dorsal pattern of expression as *Lmx1b* during limb development and it is very likely that it shares in the *LARM cis*-regulation.

Interestingly, apDV, one of the enhancers of *Apterous* (*ap*), the *Drosophila* homologue of *Lmx1b*, is maintained by a positive autoregulatory loop, albeit indirectly through the ap targets vestigial and scalloped (Vg/Sd)[22]. This suggests that positive autoregulation of *Lmx1b* is a conserved mechanism. In *Drosophila*, the apDV enhancer can only be active after ap induction by another early enhancer (apE). In the murine model, our results indicate that the LARM enhancers are not necessary for the initial activation of *Lmx1b* in the limb, but rather for its subsequent amplification above threshold levels. We also note the presence of another potential CRM 10 kb upstream of *LARM2* (Fig. 1a and Supplementary Fig. 2, asterisk) as a candidate for early Lmx1b induction that does not appear to require Lmx1b binding (i.e., it was not bound by Lmx1b in a Lmx1b-ChIP-seq at e12.5)[8]. Certainly, the presence of additional enhancers in the *Lmx1b* locus merits further investigation.

**LARM activity is required for limb-specific *Lmx1b* amplification.** To determine their functional role in *Lmx1b* regulation, we deleted the *LARM* region by CRISPR-Cas9. Mice homozygous for a 7.6 kb deletion encompassing both *LARM1* and *LARM2* (ΔLARM1/2) exhibit a limb phenotype similar to that observed in the absence of functional *Lmx1b*[3] displaying a loss of limb dorsalization, i.e., distal ventral-ventral limbs that involve the skeleton, muscles and integument (Figs. 3 and Supplementary Fig. 3). Micro-computed tomography (microCT) demonstrates biventral

distal skeletal elements (Fig. 3b–b' for the hindlimb and Supplementary Fig. 3 for the forelimb) with dorsoventral symmetrical distal phalanges (Fig. 3c–c' and Supplementary Fig. 3), sesamoid bones (Fig. 3d–d and Supplementary Fig. 3) and tali (Fig. 3e–e'). In addition, the patella, the dorsal most structure of the knee, is absent (also a notable feature in *Lmx1b* KO mice and NPS patients) (Fig. 3f–f'). These skeletal abnormalities are accompanied by corresponding muscular abnormalities (Fig. 3g–g').

In addition to the loss of limb dorsalization, the development of the cerebellum, kidney, and eye is also affected[3] in *Lmx1b* KO mice, and mutant mice die shortly after birth. However, homozygous ΔLARM1/2 mice are viable, and the organs affected in *Lmx1b* KO mice appear normal in the absence of the *LARM* region, indicating the limb-specific function of these two enhancers (Supplementary Fig. 4). The analysis of *Lmx1b* expression in ΔLARM1/2 embryos by whole mount in situ hybridization shows a normal pattern except in the limb where it was below detection limits (Fig. 3h). Analysis of limb *Lmx1b* RNA by RT-qPCR at e12.5 demonstrates a significant decrease of 60% in the steady state level of *Lmx1b* mRNA compared to normal mice (Fig. 3i). As mentioned above, the persistent expression of *Lmx1b*, albeit at a lower level in the ΔLARM1/2 mice, suggests that additional CRMs may be involved in the induction of *Lmx1b*, while *LARM1/2* amplify *Lmx1b* to levels adequate to accomplish dorsalization. Indeed a potential CRM is present 10 kb upstream of *LARM2* (Fig. 1a and Supplementary Fig. 2, asterisk), does not appear to bind Lmx1b (it was not identified by Lmx1b ChIP-seq analysis)[8], but overlaps with several chromatin-associated marks indicative of active regulation (Supplementary Fig. 2), and is worthy of further investigation. Collectively, our results establish

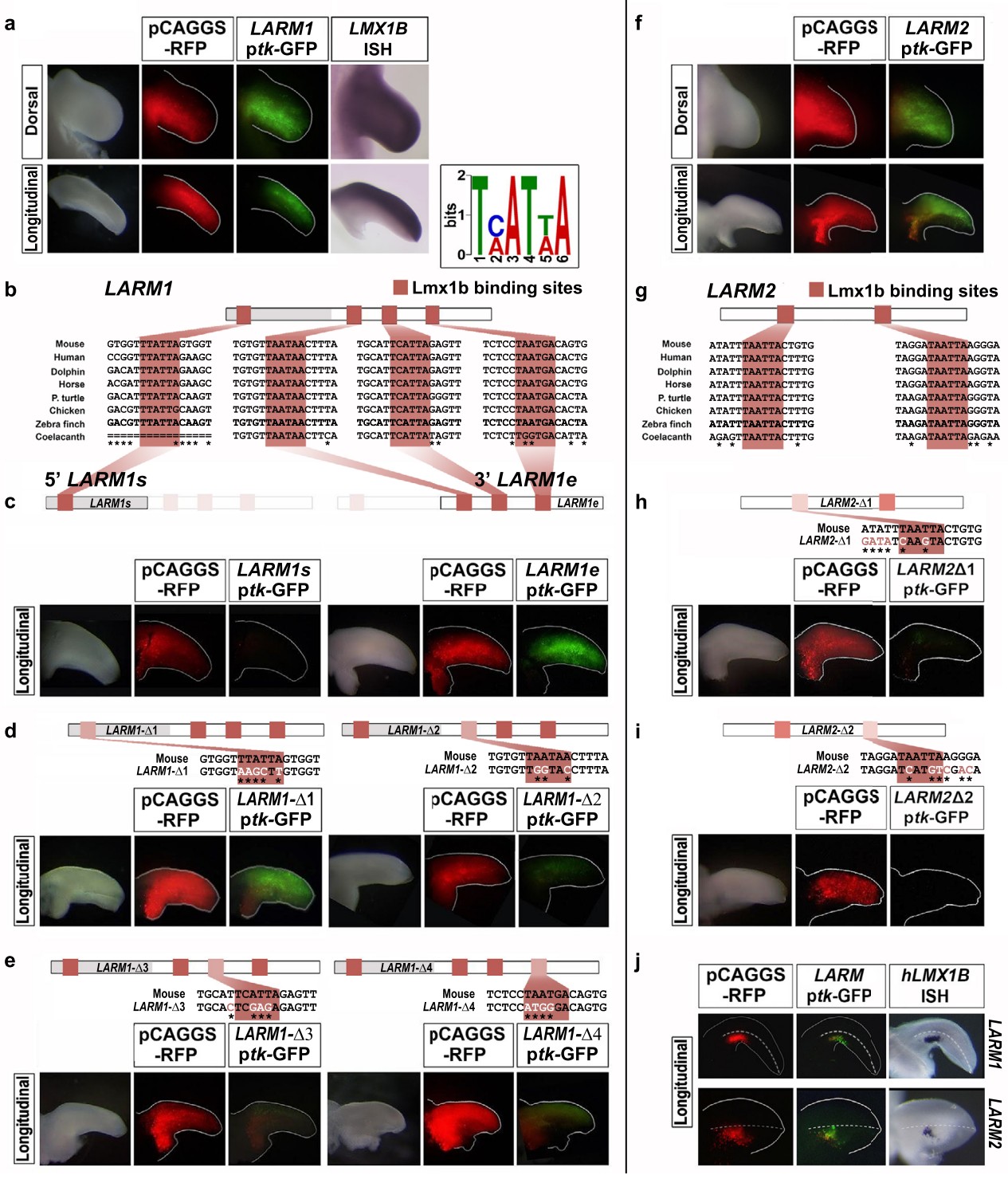

LARM1/2 as limb-specific Lmx1b autoregulatory CRMs that are necessary for normal Lmx1b transcription levels.

**Human LARM1 and LARM2 activity and role in NPS.** *LARM1/2* are conserved in humans, including the LMX1B binding sites (Figs. 1 and 2). We isolated the *hLARM* sequences and demonstrated dorsally restricted activity in the chick bioassay, either isolated or together (Fig. 4a). We also evaluated the *hLARM* sequences in transgenic mice at e12.5 (Fig. 4b). The *LARM* transgenes displayed a limb-restricted and dorsally accentuated activity. *LARM1* exhibited accentuated activity in the dorsal limb mesoderm, but weaker activity was also evident in the distal ventral aspect. *LARM2* was tightly restricted to the dorsal mesoderm but also lacked activity in the fifth digital ray (Fig. 4b, arrow). The transgene including the entire *LARM* region had

**Fig. 2 The Lmx1b binding sites are necessary for dorsal *LARM1* and *LARM2* activity.** *LARM1* (**a**–**e** and **j**) and *LARM2* (**f**–**j**) reporter activity in chick wing buds 48 hrs after electroporation. Each assay/experiment includes: bright-field view of the electroporated limbs, RFP fluorescence image (red) reflecting transfection efficiency, and GFP fluorescent image (green) showing enhancer activity. Longitudinal views illustrate activity along the dorsoventral axis (dorsal on top). **a** *LARM1* activity is restricted to the dorsal mesoderm ($n = 22$) coincident with *Lmx1b* expression (*Lmx1b* ISH for comparison). Inset showing the TMATWA consensus DNA binding motif for Lmx1b. **b** Conserved Lmx1b binding sites (LBS) are shaded clay-red in *LARM1* schematics and sequences. An asterisk indicates sequence variations across species. **c** Analysis of the isolated 5′*LARM1* element (*LARM1s*) does not convey enhancer activity ($n = 4$), while the isolated 3′*LARM1* element (*LARM1e*) is active in both dorsal and ventral mesoderm ($n = 16$). **d** Left panel. Site-directed mutagenesis of the 5′ *LARM1s* LBS in the full *LARM1* construct (*LARM1-Δ1*, $n = 5$) expands the activity into the ventral mesoderm indicating that the LBS is necessary for restriction of dorsal activity (three different site-directed mutants were generated to ensure that a new permissive/gain of function LBS was not created— Supplementary Fig. 1). **d** Right panel and **e** Disruption of any of the LBS in the 3′ *LARM1e* leads to a marked reduction in enhancer activity (*LARM1-Δ2*, $n = 7$; *LARM1-Δ3*, $n = 5$; *LARM1-Δ4*, $n = 5$). **f** *LARM2* activity is restricted to the dorsal mesoderm ($n = 13$) coincident with *LMX1B* expression (shown in (**a**)). **g** Two highly conserved LBS are present in *LARM*2 (shaded clay-red as in (**b**)). **h, i** Disruption of either LBS leads to a loss in *LARM2* activity (LARM2-Δ1, $n = 5$; LARM2- Δ2, $n = 6$). Nucleotides altered by site-directed mutagenesis are indicated in white and by an asterisk below the sequence. Dorsal or longitudinal views of the limbs are indicated on the left. **j** Ectopic expression of human LMX1B in the ventral mesoderm drives activity of the co-transfected *LARM1* or *LARM2*-reporter constructs (*LARM1* $n = 4$; *LARM2* $n = 4$). The human *LMX1B* probe used to demonstrate *LMX1B* expression by in situ hybridization does not cross-react with the dorsal expression of chicken *LMX1B*.

dorsally restricted expression with the exception of a small ventral patch in the presumptive carpal/tarsal region (Fig. 4b, arrowhead), suggesting a cooperative implementation of *LARM1* and *LARM2* activities for the refinement of dorsal restriction/enhancement.

We also explored the *LARM* region in 11 unrelated patients affected with NPS lacking sequence or copy number variation of the *LMX1B* coding region. Five of these patients were reported in a recent study[9]. In one proband (IV-7), we identified a 4.5 kb heterozygous deletion (Decipher database ID#433715) encompassing all of *LARM2* and an adjacent downstream region (Family 53 from Ghoumid et al.)[9] (Fig. 5a, b). We found that the *LARM* deletion segregates in one affected cousin (IV-1) and inferred from that result that two other affected individuals are obligate carriers (III-1 and III-4). Remarkably, individuals from this family exhibit nail dysplasia and patella hypoplasia, without ocular or renal involvement (Fig. 5c–g and k–n individual IV-7 and Fig. 5h–j individual IV-1). The nail defects were predominant on the first and second rays (koïlonychia affecting thumbs and index fingers, longitudinal striations affecting halluces) for the two individuals described, but the 5th rays were also mildly affected (nail hypoplasia of 5th fingers and toes) (Fig. 5c–j). The phenotype in this family is limb-restricted consistent with loss of enhancer activity within the *LARM* region. To investigate the phenotypic effects of the *LARM* deletion in this family, we generated by CRISPR-Cas9 a mouse model that replicated the 4.5 kb deletion carried by proband IV-7. Mice lacking this *LARM* region, termed ΔLARM2, display a ventral-ventral limb phenotype restricted to the anterior half of the limb that is clearly evident grossly (Fig. 6a). Dorsal duplication of footpads, sesamoid bones, and distal phalanx, together with absent or hypoplastic nails and hairs, was observed in the anterior digits (1–3), while the posterior digits (4 and 5) were normal (Fig. 6a, c). Consistent with the phenotype, *Lmx1b* expression was markedly reduced in the anterior half of the limb, whereas it appeared normal in the posterior half (Fig. 6a). Intrigued by the spatial restriction of *LARM2* activity and the ΔLARM2 phenotype, we generated the individual deletion of *LARM1*(ΔLARM1) and found the double ventral phenotype restricted to the posterior limb involving digits 2–5 (Fig. 6b–c). Both enhancers display no apparent anteroposterior bias in chick bioassays. In the mouse transgenic assays, only *LARM2* showed a lack of posterior activity in the fifth digit ray (Fig. 4b). Nevertheless, it is clear that, in the endogenous context, they exhibit restricted spatial modularity along the anteroposterior axis. An anteroposterior bias in humans is also commonly reported in typical NPS, where the first and second rays are usually more affected in terms of nail dysplasia[9],

concordant with the spatial activity of *LARM2*. The opposite bias has never been observed in our experience, nor reported in the literature to our knowledge.

Also, in a sporadic NPS case without ocular or renal involvement, we identified loss of heterozygosity of the *LARM* region. The *LARM*2 region was homozygous for four single nucleotide variations (SNVs) that were uncommon (each with an incidence of 16%) and one rare SNV (0.08%) (Fig. 5o). SNV-array showed several large homozygosity regions on chromosomes 9 (comprising *LMX1B*) and 17 (Fig. 5o), arguing for distant consanguinity in this individual. These results suggest that homozygosity for this rare haplotype is responsible for the autosomal recessive NPS since the heterozygous carrier parents are unaffected. Functional analyses of the *LARM2* containing the five SNVs in the chicken bioassay showed that this *LARM2* variant had reduced activity (Fig. 5p). However, a construct containing only the rare SNV within *LARM2* displayed normal activity. Therefore, either the 5 SNV are needed together, or their impact in chick is different than in human.

## Discussion

In this report, we characterize two conserved *Lmx1b*-associated *cis*-regulatory modules (*LARM1* and *LARM2*) that are bound by Lmx1b and required to amplify *Lmx1b* expression in the limb to levels sufficient to accomplish limb dorsalization. Thus, *LARM1* and *LARM2* are two limb-specific *Lmx1b* enhancers that display remarkable modular spatial activity and that are required for establishing the correct dorsoventral pattern across the anterior-posterior axis.

Consistent with being limb-specific enhancers, mice in which the *LARM* region has been removed do not develop other *Lmx1b*-associated abnormalities that might jeopardize survival, thereby offering an extraordinary opportunity to study the functional capacities of ventral-ventral limbs. Indeed, the limbs of ΔLARM1/2 mice are insufficient for locomotion. ΔLARM1/2 mice cannot walk, but rather use an undulating or irregular wiggling motion because their limbs are unable to lift their bodies to move them forward. This stresses the notion that fins capable of supporting the body weight, such as those observed in *Tiktaalik*, must have been an initial step in the transition to tetrapods[23]. Considering that animal forms likely evolved by altering the regulation of key developmental genes, modification of the *Lmx1b* landscape may have been a critical step in the acquisition of dorsoventral polarized fins capable of lifting and moving the body, a hypothesis that deserves further investigation. In contrast, the partial (or modular) dorsoventral alterations exhibited by ΔLARM1 and ΔLARM2 homozygous mice do not appreciably interfere with

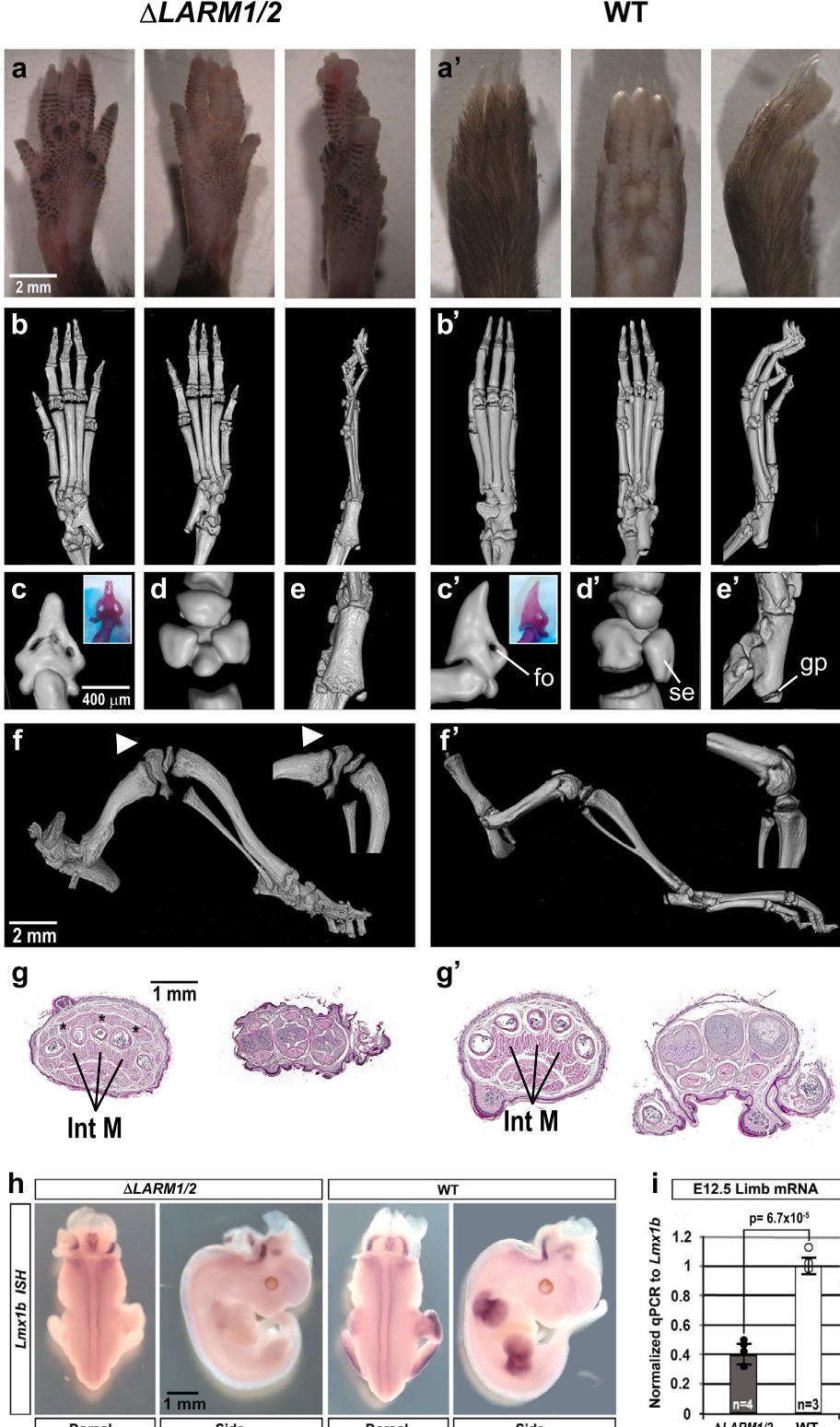

locomotion, although abnormal gait has been occasionally observed.

We also show that disruption of these enhancers can cause human pathology since loss-of-function variations in the *LARM* region are responsible for a limb-specific form of human NPS. This limited form of NPS is not associated with the typical risk of chronic kidney disease or glaucoma. Moreover, there is no

protein-coding variation. Recognition that disruption of the *LARM* region can cause this limited form of NPS provides these patients with a more accurate assessment of their condition.

The NPS phenotype is attributed to haploinsufficiency, i.e., reduced levels of LMX1B due to the loss of one allele. Our studies further characterize the pathogenicity of NPS to reduced levels of LMX1B. In one NPS family, a single allelic deletion of *LARM2*

**Fig. 3 Mice lacking the *LARM region* exhibit a double ventral limb phenotype. a-a'** Dorsal, ventral and lateral gross morphology of hindlimbs (forelimb morphology in Supplementary Fig. 3). **b-b'** microCT scan views of a 3-week-old ΔLARM1/2 homozygous mouse hindlimbs showing footpad development and the absence of nails and hair in the dorsal autopod compared to wild type (WT). **c–e** Magnified views of the digit tips (inset alizarin red staining of the distal phalanx) showing symmetrical ventral features, i.e., bony ventral foramen (fo) associated with the toe pad, symmetrical ventral sesamoid bones (se) of the metatarsal-phalangeal joint, and the ventrally oriented growth plate (gp) of the proximal tali display dorsoventral symmetry in the absence of the *LARM region*. Compare with the normal dorsoventral asymmetry of wild-type controls (**c´–e´**). In images **c–e** dorsal is to the left. **f-f'** The patella is absent in ΔLARM1/2 mice (hindlimb lateral view, white arrowheads). **g-g´** Transverse sections of the autopod show duplicated flexor tendons and intrinsic muscles (Int M) (asterisk) in the ΔLARM1/2 mouse ($n = 3$, wild type $n = 2$). **h** In situ hybridization of *Lmx1b* expression in limb buds is below detection in animals lacking the *LARM region*, while expression in the neural tube is equivalent. **i** Comparative RT-qPCR analyses of *Lmx1b* mRNA levels in the whole hindlimb bud of e12.5 ΔLARM1/2 (black dots) and WT (white dots) embryos. The level of expression of the mutant is 40% of the control (set to 1). $P = 6.7 \times 10^{-5}$ (two-tailed, unpaired t-593 test, error bars represent standard deviation) (control limb $n = 3$, mutant limbs $n = 4$). Source data for the RT-qPCR are provided as Supplementary Data 3.

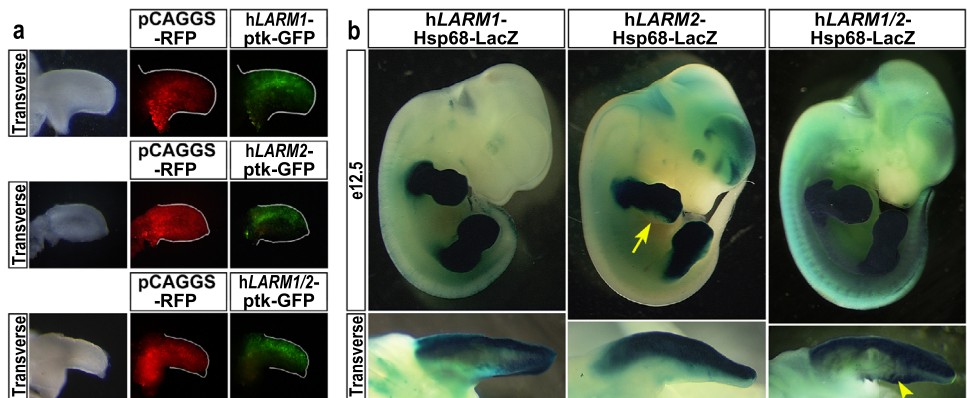

**Fig. 4 The human *LARM* region also exhibits dorsal enhancer activity. a** Human *LARM* constructs electroporated into chick wing buds display dorsally restricted expression (*hLARM1*, $n = 13$; *hLARM2*, $n = 7$; *hLARM1/2 region*, $n = 3$). **b** Similarly, transgenic mice containing the human *LARM* sequences linked to a LacZ reporter demonstrate dorsally accentuated activity. All three *LARM1* transgenic embryos display limb-restricted, dorsally accentuated activity. *LARM2* transgenic embryos exhibit tight dorsally restricted activity in the limb (5/5). LacZ staining is reduced-to-absent in the posterior distal autopod mesoderm (5th digit region, yellow arrow). Transgenic embryos containing both *hLARM1/2* (~8 kb, *hLARM1/2*) also show activity restricted to the dorsal limb (7/7). However, focal ventral activity at the zeugopod/autopod junction is also evident (yellow arrowhead).

yields limb features diagnostic of NPS (incomplete limb dorsalization) indicating that LMX1B protein levels are below the normal patterning threshold. In another family, homozygosity of a functionally impaired *LARM2* allele also yields limb features diagnostic of NPS. In both of these families, the remaining *LARM1* enhancer, which demonstrates clear activity in transgenic mice and chicken bioassays, appear able to support LMX1B amplification to partially dorsalize the posterior limb and avert a more severe ventral-ventral phenotype. During the final submission of this manuscript an additional NPS family with a confirmed deletion that removed both *LARM1* and *LARM2* in one allele was identified. Further mapping and studies are underway. Together our results point to the contribution of both allelic sets of the *LARM* enhancers to get a fully functional dose of LMX1B across the anterior-posterior axis.

Congenital limb anomalies are relatively common[24,25] with syndromic forms associated with more than a hundred genes. The association of multiple affected organs (developmental pleiotropy) provides a clue to the affected gene and permits a high diagnostic yield. However, more than half of limb anomalies are isolated without other malformations, and the diagnostic yield of genetic evaluation remains low in these cases due to, at least in part, an emphasis on evaluating coding sequences. During morphogenesis, tissue-specific CRMs cause developmental pleiotropy by regulating genes in key developmental pathways in precise temporal and spatial patterns. Thus, tissue-specific CRMs are potential candidates to explain isolated limb anomalies. Our findings, as well as others linking limb-specific CRMs to limb anomalies[26–29], support this concept. Characterization of CRM-disease associations represents a forthcoming opportunity in clinical genetics, not only for limb anomalies, but also for other isolated malformations.

## Methods

**Animal procedures**. All animal procedures were reviewed and approved by the Loma Linda University Institutional Animal Care Use Committee (IACUC) or by the Bioethics Committee of the University of Cantabria and performed according to the EU regulations, animal welfare and 3R principles. Representative images are shown, but in all animal analyses performed, at least two independent specimens, and in most cases three or more, were used to confirm the morphologic pattern.

**Patients**. We obtained informed written consent from all participants for genetic analyses. Analyses were performed on a diagnosis basis in the University hospital of Lille, following the bioethics rules of French law. The study was reviewed by the Institutional Ethics Committee of the University of Lille and was found to be in accordance with the criteria set by the Declaration of Helsinki. No identifiable images of human participants are used.

**Functional enhancer validation in chicken bioassays**. We used a thymidine kinase (*tk*) promoter-driven GFP reporter (kind gift of Masanori Uchikawa)[30] to generate enhancer constructs. Functional analyses of the *LARM1* and *LARM2* constructs were performed by electroporation into presumptive limb mesoderm of Hamburger and Hamilton stage (HH) 14 chicken embryos. Co-electroporation of a β-actin promoter-driven RFP construct (pCAGGS-RFP, kind gift from Cheryl Tickle)[31] was used to determine transfection efficiency. Electroporation was performed using the CUY21 electroporation station (Protech International, Boerne, TX). Embryos were incubated for 48 h before harvesting for visualization of GFP activity and digital image acquisition (Sony DKC-5000) into Adobe Photoshop (version 6.0, acquisition; version 2020, compilation).

To demonstrate that LMX1B could induce construct activity, pCDNA3.1-hLMX1B (kind gift from Roy Morello)[5] was co-electroporated with either p*tk*-LARM1-GFP or p*tk*-LARM2-GFP into the ventral mesoderm of stage HH23 chicken limb buds.

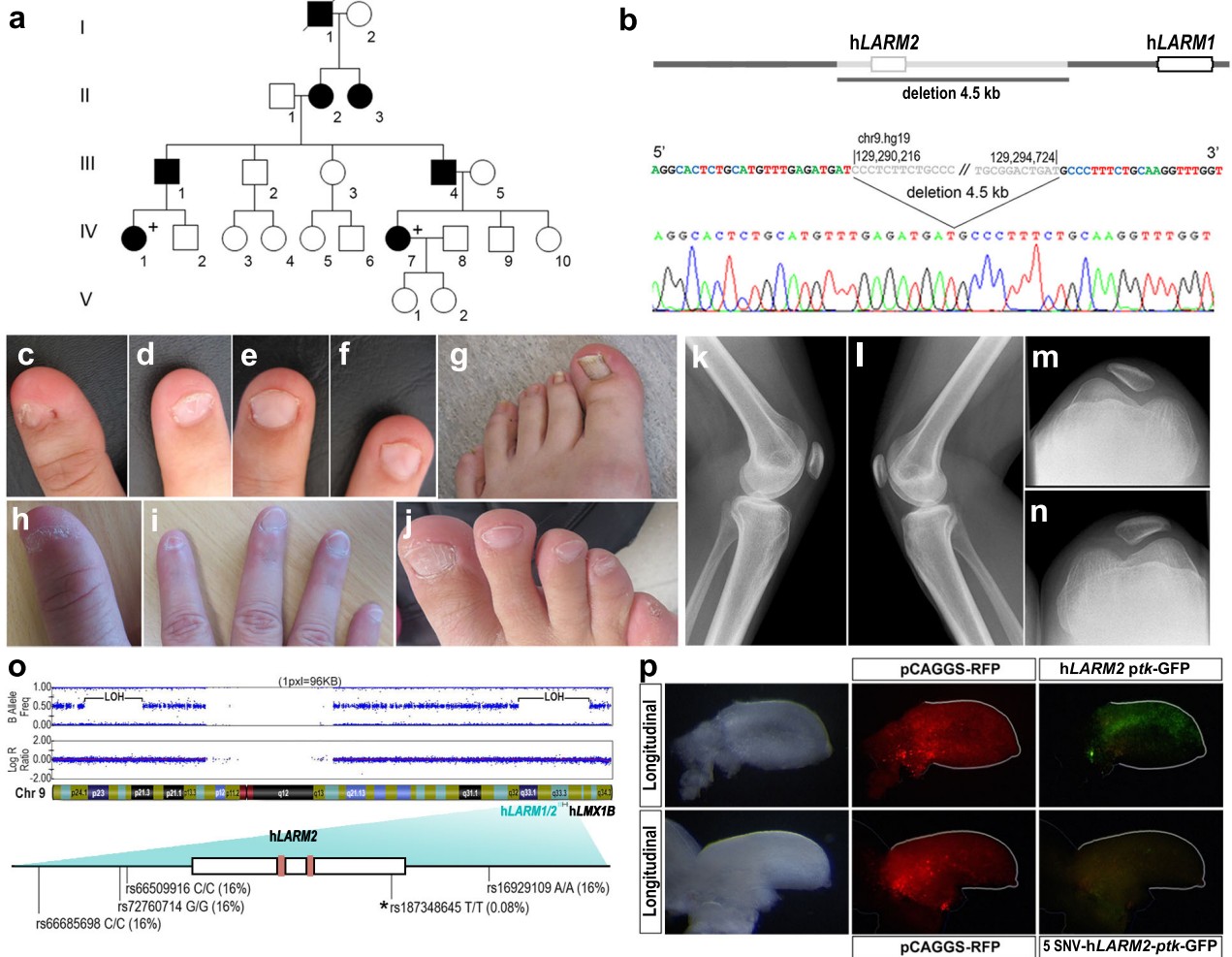

**Fig. 5 Clinical features of *LARM* loss-of-function. a** Pedigree of a family with a *LARM* deletion. **b** The 4.5 kb region deleted removes all of *LARM2*. **c–g**, **k–n** Phenotypic images of individual IV-7. **c–d** Koïlonychia of thumb and 2nd finger. **e–f** Triangular lunulae of 3rd and 4th fingers. **g** Nail dysplasia of the hallux showing longitudinal striations. **h–j** Phenotypic images of individual IV-1. **h** Koïlonychia of thumb. **i** Hypoplastic nails, ungueal dysplasia of 2nd finger. **j** Ungueal dysplasia of right foot predominating on 1st and 5th toes. **k–n** Knee X-rays showing bilateral hypoplasia of the patella. **o** Schematic of the patient's chromosome 9 showing large segments of the chromosome with loss of heterozygosity (LOH), i.e., homozygosity, when comparing the allele frequencies to the Log R ratio of the alleles. One of the homozygous regions includes the *LARM-LMX1B* locus. The homozygous h*LARM2* sequence showing the five single nucleotide variations (SNVs). The asterisk indicates the rare (0.08%) sequence in *LARM2*. The LMX1B binding sites are indicated as clay-red boxes. **p** Using site-directed mutagenesis, we generated a human *LARM2* construct containing the patient's 5 SNVs; following electroporation into embryonic chick wings, the patient-*LARM2* sequence showed markedly reduced activity (*n* = 6; compare with the activity of the common h*LARM2* sequence in Fig. 4a). Interestingly, mutation of only the single SNV within the conserved *LARM2* region did not alter *LARM2* activity.

**Cloning and site-directed mutagenesis**. Primers used for the isolation of enhancer sequences from genomic DNA are listed in Supplementary Data 1. Disruption of the Lmxb1 binding sites was performed using the QuikChange Lightning Site-Directed Mutagenesis Kit (Agilent Technologies, Santa Clara, CA) following manufacturer recommendations and confirmed by Sanger sequencing. Briefly, nucleotides were modified to disrupt the binding site with a change of at least 3 nucleotides, not add another binding site, and add a restriction enzyme site for evaluation of successful mutagenesis. All potential binding site changes were evaluated by AliBaba2.1[32] and/or TRANSFAC®[33] prior to construction to ensure that no new binding sites present in the limb were introduced.

**In vivo transgenic reporter assays**. *Lmx1b*-associated regulatory modules were isolated from human genomic DNA with the primers listed Supplementary Data 1 (a Microsoft Excel file) and cloned into the hsp68-LacZ kindly provided by Dr. Ahituv[34]. The constructs were used to generate transgenic embryos (Cyagen transgenic service, Santa Clara, California). The embryos were harvested at e12.5 and processed for detection of LacZ activity.

**Analyses of published data**. Limb ChIP-seq data were obtained from the Gene Expression Omnibus database (GEO, http://www.ncbi.nlm.nih.gov/geo/) under the accession numbers GSE84064 for Lmx1b[8], GSE42413 for H3K27Ac[11], GSE13845

for p300[10], and GSE42237 for both H3K27me and H3K4me[12]. RNA Pol II and Med12 ChIP-seq data were available from Berlivet and coworkers[13]. Previously published data containing the genomic coordinates of interest were uploaded to the UCSC genome browser and converted to the mouse build mm10 using the liftover tool. Capture-C data were mined from previous published work deposited in the GEO database under accession number GSM2251518[19].

**CRISPR-Cas9 mediated enhancer knockout mice generation**. The knockout mouse strain for the *LARM* region (Δ*LARM1/2*) was generated with the use of CRISPR-Cas9. Single guides RNAs (sgRNA) flanking the *LARM* locus listed in Supplementary Data 1 were designed using breaking-Cas[35]. The sgRNAs were generated (Sigma-Aldrich) and cleavage efficiency tested by Sigma-Aldrich using their cel-1 assay in mouse neuroblastema (N2a) cells with the primer pairs listed in Supplementary Data 1. Generation of knockout mice was performed at the National Center of Biotechnology's transgenic core facility (CNB-CSIC). Following cytoplasmic injection of 50 ng/μl each sgRNA and 100 ng/μl of Cas9 mRNA, in injection buffer (1 mM Tris–HCl pH 7.5; 0.1 mM EDTA pH 7.5) into B6CBA mouse strain single cell embryos, the treated embryos were placed in donor CD1 pseudopregnant females (mated with vasectomized CD1 males to induce pseudopregnancy). Genotyping of founder (F0) mice for the identification of the desired deletion was performed with the use of the Phusion high fidelity polymerase (Thermo Scientific) with the primers listed in Supplementary Data 1. A founder female B6CBA was bred to

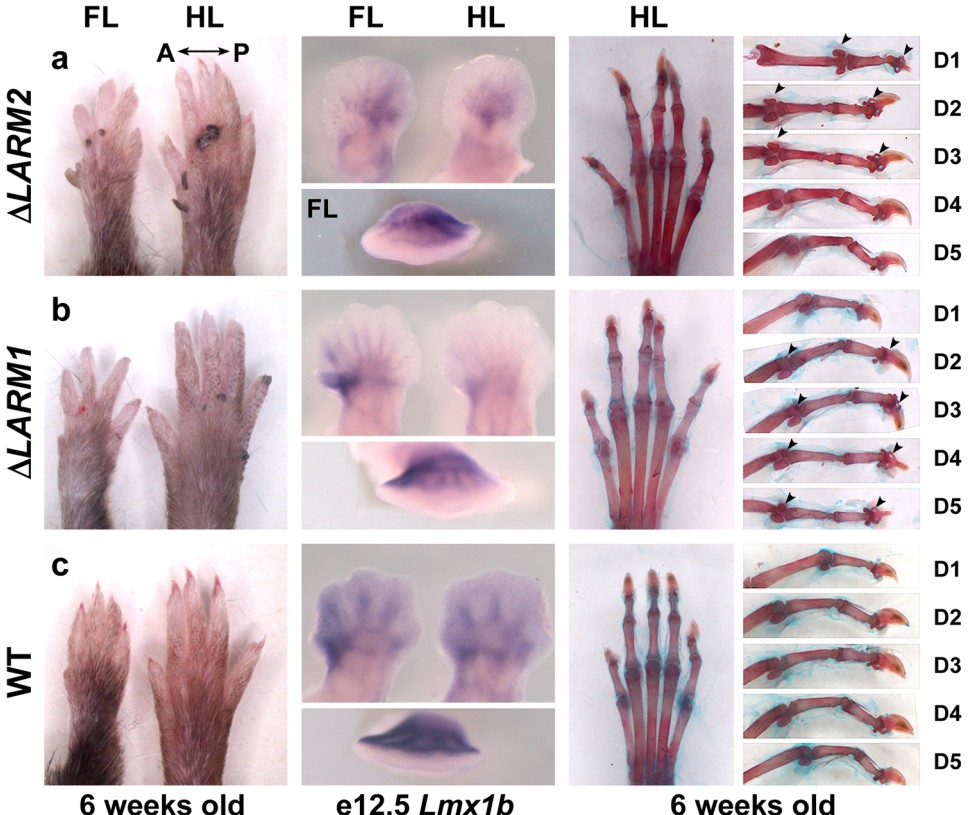

**Fig. 6 Individual deletions of *LARM*1 and *LARM*2 reveal posterior and anterior contributions, respectively, to dorsalization. a** Homozygous Δ*LARM*2 mouse forelimb (FL) and hindlimb (HL) autopods showing loss of dosalization exclusively in the anterior aspect at 6 weeks (left), reduced anterior expression of *Lmx1b* in e12.5 limb buds (middle) and ventral-ventral morphology specifically in anterior 1–3 digits (right). **b** Homozygous Δ*LARM*1 mouse showing loss of dorsalization that is accentuated posteriorly at 6 weeks (left), reduced posterior expression of *Lmx1b* in e12.5 limb buds (middle) and ventral-ventral morphology specifically in digits 2–5 (right). Note the progressive loss of dorsalization towards digit 5 (e.g., increasing size of duplicated ventral sesamoid bones). **c** Equivalent views of a wild-type littermate for comparison. Arrowheads highlight the duplicated ventral sesamoid bones and distal phalanges.

C57/BL6 for the generation of the *LARM1/2* knockout mice strain after performing Sanger sequencing of the PCR amplicons that confirmed a 7,564 bp deletion encompassing the *LARM* locus (Chr2: 33555354-33563008; Mouse July 2007(NCBI/mm9) assembly). The genotyping strategy included Sanger sequencing of the PCR amplicons for the off-target regions providing the highest score for each of the sgRNAs according to Breaking-Cas with the use of the primers listed in Supplementary Data 1 confirmed no genetic alteration.

The gRNAs used to generate the Δ*LARM1* and Δ*LARM2* lines were designed with CHOPCHOP (https://chopchop.cbu.uib.no/) and are listed in Supplementary Data 1. The mutant lines were generated by electroporation using the Alt-R CRISPR-Cas9 System from IDT (https://eu.idtdna.com/pages/products/crispr-genome-editing/alt-r-crispr-cas9-system) and the NEPA21 electroporator and the CUY-1.5 electrode following manufacturers recommendations[36]. B6CBAF1/J fertilized eggs were collected from the oviducts of e0.5 pregnant females. The collected eggs cultured in WM medium were washed with Opti-MEM (Gibco; 31985-047) three times to remove the serum containing medium. The eggs were then lined up in the electrode gap filled with the electroporation solution, electroporated, and transferred into pseudopregnant foster mice[36] at the transgenesis service of the Instituto de Biotecnologia y Biomedicine de Cantabria IBBTEC, Spain. A 2,818 bp and 5,266 bp deletion for *LARM1* (chr2: 33555387-33558204; Mouse July 2007(NCBI/mm9) assembly) and *LARM2* (chr2: 33558386-33563651), respectively, was confirmed by sanger sequencing.

**MicroCT**. Three-week-old mouse limbs were scanned with Skyscan1172 at 40 kV, 100 μA, and 27.03 μm pixel resolution. Subsequent reconstruction was performed using the NRecon reconstruction software (Ver 1.6.10.2) and compiled with the CTvox version 3.3.1volume rendering software.

**Skeletal preparations**. After removal of the skin and viscera, animals were fixed in 95% ethanol. Alizarin red and alcian blue skeletal staining was performed following standard procedures.

**Histology**. Animals were subjected to intravascular perfusion of 4%PFA with the use of a peristaltic bomb. Gross morphologic and histologic analyses were performed on the limbs, skull, brainstem, kidneys, and eyes. The soft tissues (brain, kidneys and eyes) were fixed in 10% phosphate-buffered formalin, while the limbs were decalcified then post-fixed with 10% phosphate-buffered formalin. The tissues were paraffin-embedded following standard procedures and stained with hematoxylin and eosin.

**RT-qPCR**. The hindlimbs from e12.5 wild type and Δ*LARM1/2* homozygous embryos were dissected out in cold RNase-free 1X phosphate-buffered saline (PBS) pH 7.4. Total RNA was extracted with RNeasy Plus Mini Kit (Qiagen) and 500 ng of total RNA was reverse transcribed to produce first-strand cDNA with iScriptTM cDNA Synthesis kit (Bio-Rad) using standard conditions. RT-qPCR was carried out on an Applied Biosystems StepOnePlus using NZYSpeedy qPCR Green Master Mix, ROX plus (NZYTech). The primers used to amplify *Lmx1b* were forward (Fwd), GAGCAAAGATGAAGAAGCTGGC[37], and reverse (Rev), GGCCACGA TCTGCTGCTG. Relative transcript levels were normalized to GAPDH (Fwd, TGCAGTGGCAAAGTGGAGAT; Rev, ACTGTGCCGTTGAATTTGCC). Three-four biological replicates were analyzed for each genotype, with 3 technical replicates per sample. The expression levels of mutant samples were calculated relative to wild-type controls (average set to 1). Significance of differences were determined using the two-tailed, unpaired *t*-test and reported with standard deviation error bars.

**In situ hybridization**. Whole-mount in situ hybridization was performed following standard procedures[38] using digoxigenin-labeled antisense RNA probes for mouse *Lmx1b*[3] and Human *LMX1B*[5]. Briefly, embryos were harvested in cold RNase-free 1× PBS (pH 7.4), fixed overnight cold (4 °C) MEMFA (100 mM MOPS (pH 7.4), 2 mM EGTA, 1 mM MgSO4, 3.7% (v/v) formaldehyde) and post-fixed overnight in 90% methanol at −20 °C. The embryos are then rehydrated in graded alcohols/PBS + .0.1% Tween, treated with protease K (10 mg/ml) for 20 minutes, rinsed in 0.1 M triethanolamine (TEA), and acetylated

(0.25% Acetic Acid in 0.1 M TEA). The embryos are then fixed in 4% paraformaldehyde in PBS + 0.1% tween (PBT), rinsed in 0.1 M glycine (in PBT) and then rinsed in hybridization mix (50% formamide, 5× saline-sodium citrate buffer (SSC), 1 mg/ml Bakers yeast RNA, 100 μg/ml heparin, 1× Denhardt's solution, 0.1% Tween 20, and 0.1% CHAPS) for 1 h. The embryos are then treated with fresh hybridization mix containing probe (0.2 μg/mL), incubated overnight (at least 14 hrs) at 60 °C, then rinsed with multiple FSC (50% formamide, 2× SSC, 0.1% CHAPS, 50 mM glycine) washes at 63 °C, detected using anti- digoxigenin antibodies conjugated to alkaline phosphatase (Sigma/Aldrich) in 2% blocking reagent (Boehringer)/PBT, and colorize with the BCIP/NBT substrate in a CT salt mix (150 mM NaCl, 25 mM MgCl$_2$) for 20–60 min.

The colorized embryos were imaged using Leica MZ8 dissecting microscope with attached Sony DKC-5000 camera into Adobe Photoshop (version 6.0, acquisition; version 2020, compilation). To generate the chicken *Lmx1b* probe we isolated a 729 bp fragment from chicken cDNA that spans 5 introns (exons 2– 7) of the *Lmx1b* gene using the following primer pairs: FWD: 5′ GGATCGCT TTCTGATGAGG 3′, REV: 5′ GATGTCATCATTCCTTCCATTCG 3′. The isolated fragment was cloned into pCRII-TOPO vector with dual Sp6 and T7 promoters for in vitro transcription (ThermoFisher Scientific) and sequence verified by Sanger sequencing.

**LARM screening in NPS patients**. DNA from patients was extracted from blood according to standard methods. *LARM1* and *LARM2* were sequenced on an ABI Prism 3730XL Genetic Analyzer (Applied Biosystems, Courtaboeuf, France) using Big Dye Terminator v3.1 Cycle Sequencing Kit, after PCR amplification using the following primers pairs:
hLARM1_5p1-F GTGTAGGTTTGACGGTGGGATTTTCC,
hLARM1_3p1-R GCTGGAGCCCATGAGAAGATTGC,
hLARM2_5p1-F CCCACGGCAGGAGTTATAAGCAAGG,
hLARM2_3p1-R CGGACCAGGAGAAACATTCTTCTGTG.

Copy number of *LARM1* and *LARM2* was studied by real-time quantitative PCR using SYBR Green technology (Applied Biosystems®, Saint Aubin, France) with the following primers pairs: LARM1-F AATTAACGGCTCCTCCCTG, LARM1-R GCCTTCTTCCTACTTCTGTCA, LARM2-F GTCTCTGCC CCTCGCTGA, LARM2-R CGTGGGCAATATGGCTTTGAA. Quantification of the target sequences was normalized to an assay from *RPH Polymerase* (NR_002312), and the relative copy number was determined on the basis of the comparative ΔΔCt method using a normal control DNA as the calibrator.

**Reporting summary**. Further information on experimental design is available in the Nature Research Reporting Summary linked to this paper.

## Data availability

The data that support the findings of this study are available from the corresponding author upon reasonable request. The copy number variation data (4.5 kb heterozygous deletion) described in this study has been reported in the Decipher database under the ID#433715 (https://www.dechipergenomics.org/). Publically available Lmx1b gene array data of e12.5 limbs analyzed in this study is available through the Gene Expression Omnibus (GEO) database, under accession number GSE34732. Published limb ChIP-seq data are available under the GEO database accession numbers: GSE84064 for Lmx1b, GSE42413 for H3K27Ac, GSE13845 for p300, GSE42237 for both H3K27me & H3K4me. RNA Pol II and Med12 ChIP-Seq datasets are available as Supplementary datasets 1 and 2 from Berlivet and coworkers[13]. Capture-C data was mined from previous published work deposited in the GEO database accession number GSM2251518. Source data are provided with this paper. The data generated in Fig. 3I of this study are provided as "Source Data for RT-qPCR of dLARM1/2 and WT hindlmbs". Additional embryos from the enhancer assay data represented in Fig. 2 and the transgenic data shown in Fig. 4 are provided in the supplementary data, Supplementary Data 2.

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

## Acknowledgements
The authors are grateful to Dr Boris Keren for the analysis of the SNP array in family 2 and to Laura Galán for excellent technical support. This work was supported in part by grants from the Spanish Ministerio de Ciencia, Innovación y Universidades (M.A.R) (BFU2017-88265-P); the National Organization for Rare Disorders (K.C.O.), and the Loma Linda University Pathology Research Endowment Fund (K.C.O.).

## Author contributions
E.H., F.P., M.A.R. and K.C.O conceived and designed the project. E.H., C.U.P, C.D.S., L.A.I., A.L.G. and A.N. acquired and analyzed murine and human *LARM1/2* characterization studies in chickens. E.H., C.U.P., A.L.G. and K.C.O. acquired and analyzed human *LARM1/2* studies in transgenic mice. E.H., S.L., K.C.O. and M.A.R. generated and analyzed the CRISPR-cas9 knockout model mice. F.P. and S.M.H. supervised the acquisition and analysis of the human Nail-Patella Syndrome (NPS) data. F.E., A.-S.J., F.F., J.-M.G. and C.F. acquired and analyzed human NPS data. E.H., F.P., K.C.O. and M.A.R. drafted the manuscript. All authors were involved in editing/revising the manuscript.

## Competing interests
The authors declare no competing interests.
