## [Peer Review File · Nature Communications]

Reviewers' comments:

Reviewer #1 (Remarks to the Author):

The manuscript by Haro et al. builds on early microarray analysis identifying genes regulated by Lmx1 and a subsequent study using ChIP to explore regulatory elements directly bound by the transcription factor. This led to the description of two conserved elements up-stream of Lmx1 (LARM1 and Larm2) which were good candidates for playing a role in the auto regulation of Lmx1 transcription. Much of this manuscript explores the manner of this autoregulation and the role of these elements in recapitulating the ventralised phenotype of the limbs observed in the Lmx1 knockout mouse. Two families with aspects of the limb defects of nail-patella syndrome (but none of the associated phenotypes) are subsequently identified with deletion of /point mutations in one these elements. In combination, these experiments nicely link the identification of tissue specific enhancers to abnormal development and to disease pathology. There is however a slight disconnect between the animal experiments focusing mainly on autoregulation and the human mutations identified.

There's always a fear when making mutations in binding sites that new sites are created instead and here each site seems to have been mutated differently. Can the authors discuss how they decided on which mutation to make? Given that most changes seem to inactivate the enhancer and, at least in the chick electroporation assay, only one up-regulates it would be good to see this increase in expression recapitulated with a second different mutation which inactivates in a different context.

The ventral turn on expression driven by LARM2 responding to ectopic expression of hLMX1B is a lovely confirmation that autoregulation, presumably via the identified sites, plays a part. However, the same experiment using the LARM1 construct is clearly missing.

Of the 5 SNVs identified in the nail-patella syndrome patient, it looks as if only one is actually within hLARM2- it seems important to show what effect just this one SNV has on expression.

The chromatin marks in figure 1 suggest that Larm1 is the much more 'convincing' enhancer carrying (more) p300 and RNA pol2 but the human data implicates LARM2- Are both actually required? What happens to the limb phenotype when either is knocked out individually in the mouse?

Minor points

Given that most of the data in figure 1 comes from published studies (identified in the M&Ms), it would be good to get matching references into the text.

In figure 2, the brick red shading of the site in the delta4 construct is missing the final base. And in the figure legend, I think you've switched 3' and 5' in C).

In Figure 3- would be worth pointing out these are all hindlimbs. Good to include some fore limb pictures too?

There's a discrepancy between the primer sequences in TableS1 and figure 2 – a few seem to have been switched and something's happened to the delta1 sequence (reversed?). Please fix.

Table S1- p2 – instead of primer sequences the text reads 'in the paper there are two sets of primers- were both used?' Please correct.

In the M&M- in situ hybridisation reads 'To generate the chicken probe....' Please add the missing text.

Reviewer #3 (Remarks to the Author):

The manuscript by Haro et al. studies the cis-regulation of LMX1B, an essential gene for the proper establishment of the dorso-ventral axis in developing limbs. Building on previous research, the authors identify and dissect molecularly two upstream enhancers, termed LARM1 and LARM2, both directly bound by the Lmx1b protein. Using a chicken GFP-reporter system and through site-directed mutagenesis, the authors demonstrate the dorsally-restricted activity of the two regulatory elements, as well as the importance of Lmx1b-binding sites in driving enhancer function. Their findings are supported by the in vivo deletion of the enhancer region, causing a limb-restricted phenotype that recapitulates tissue-specific aspects of the full Lmx1b knockout. The study is nicely

rounded with the identification of two unrelated human cases carrying mutations affecting the enhancer elements and displaying limb-restricted phenotypes.

This beautiful and important piece of work highlights the modular nature of regulatory elements in controlling tissue-specific aspects of gene expression and in building up complex developmental expression patterns. The manuscript is well written, clear, concise and with an appropriate interpretation and discussion of the data. For such reasons, I fully support its publication in Nature Communications.

I also provide suggestions for improvement that the authors might take into consideration:

Minor comments

1- In FigureS1, , it is difficult to appreciate the overall TAD structure of the locus because of the low resolution of the Hi-C data as well as due to the region that was chosen to display.

I would recommend the inclusion of higher resolution data in the panel (Bonev et al 2017, Cell). In addition, the displayed region might be extended at least 1 Mb downstream (from 34,500,000 to 32,500,000) to fully appreciate the extension of the Lmx1b TAD.

2- The qPCR results on LARM KO mouse limbs shows residual expression of Lmx1b. This suggests that, although the Lmx1b enhancers are required to sustain limb expression levels, either the maintenance or initiation of expression is controlled by additional elements. Are there other elements near Lmx1b that could be performing such action? This aspect might be discussed briefly in the manuscript.

From Capture-C data, seems that Lmx1b interactions extends downstream of the gene. Figure 1 should be adapted accordingly to show the downstream interacting region and p300 and H3K27Ac CHIP-Seq tracks might be included in the upper panel, to display potential regulatory elements.

3- Lines 50-52. The assumption that haploinsufficiency accounts for the syndromic features is better sustained by the overlapping symptoms between missense, nonsense, or deletion mutations (Dunston et al., 2004, Genomics) or by in vitro experiments (Dreyer et al., 2000, Hum Mol Genet),

rather than by interspecies differences as stated by the authors. I would reformulate this statement and cite these studies, to be more precise

4- Lines 183-185. This is a very interesting evolutionary hypothesis that might be worth discussing in more detail.

The LARM region is partially conserved in the fin-lobbed fish *Coleacanth*, a representative species of the transition from fins to limbs. Is this enhancer region also partially conserved in ray-finned fishes?

Figure 2 would benefit from the inclusion of genomic sequences of such fishes in the corresponding panels. In addition, it would be advisable to comment on the genomic differences (and the link with fin/limb morphology) between ray-finned fishes, lobe-finned fishes and tetrapods.

5- It is unclear what marks the asterisk in Figure 1.

6- In the legend of Figure S1, it is written "topographically" instead of "topologically"

Darío G. Lupiáñez

Identification of limb-specific Lmx1b auto-regulatory modules with Nail-Patella Syndrome pathogenicity.

Endika Haro^{1,2}, Florence Petit^{3,4}, Charmaine U. Pira¹, Conor D. Spady¹, Sara Lucas-Toca², Lauren A. Ivey¹, Austin L. Gray¹, Fabienne Escande^{4,5}, Anne-Sophie Jourdain^{4,5}, Andy Nguyen¹, Florence Fellmann⁶, Jean-Marc Good⁶, Christine Francannet⁷, Sylvie Manouvrier-Hanu^{3,4}, Marian A. Ros^{2*}, Kerby C. Oberg^{1*}

Response to Reviewer's comments

We would like to thank the referees for their constructive insights and helpful comments towards the improvement of our manuscript.

We are indeed grateful as we feel the newly acquired data from the requested additional experiments provide further support for the role of the *LARM* region in Lmx1b autoregulation and an interesting twist to the regulation of Lmx1b, that of modular (anterior-posterior) control. Below is a specific point-by-point response to the reviewer's questions and comments. For the sake of clarity, we have reproduced the comments of the reviewers and have introduced our answers in blue.

Reviewers' comments:

Reviewer #1 (Remarks to the Author):

The manuscript by Haro et al. builds on early microarray analysis identifying genes regulated by Lmx1b and a subsequent study using Chip to explore regulatory elements directly bound by the transcription factor. This led to the description of two conserved elements up-stream of Lmx1b (LARM1 and Larm2) which were good candidates for playing a role in the auto regulation of Lmx1b transcription. Much of this manuscript explores the manner of this autoregulation and the role of these elements in recapitulating the ventralised phenotype of the limbs observed in the Lmx1b knockout mouse. Two families with aspects of the limb defects of nail-patella syndrome (but none of the associated phenotypes) are subsequently identified with deletion of /point mutations in one these elements. In combination, these experiments nicely link the identification of tissue specific enhancers to abnormal development and to disease pathology. There is however a slight disconnect between the animal experiments focusing mainly on autoregulation and the human mutations identified.

1. There's always a fear when making mutations in binding sites that new sites are created instead and here each site seems to have been mutated differently. Can the authors discuss how they decided on which mutation to make?

We agree with the referee in this statement. We now explain in the Method section (page 22) how these mutations were designed. Briefly, nucleotides were modified to disrupt the binding site with a change of at least 3 nucleotides, not add another binding site, and add a restriction enzyme site for evaluation of successful mutagenesis. Binding site changes were evaluated by AliBaba^{2.129} and/or TRANSFAC^{®30} prior to construction to ensure that no new potential TF binding sites were introduced. If a new binding site is introduced and cannot be avoided, we check to see if the associated transcription factor is expressed within the limb during development. This has been added to the M&M section (pages 23 and 24, lines 473-477)

2. Given that most changes seem to inactivate the enhancer and, at least in the chick electroporation assay, only one up-regulates it would be good to see this increase in expression recapitulated with a second different mutation which inactivates in a different context.

Following the referee's recommendation, we generated two additional mutations of the Lmx1b binding site in the *LARM1* silencer, the only one that resulted in an upregulation of the enhancer activity. These two additional mutations also resulted in ectopic ventral activity making it very unlikely that it was the result of the creation of a new site. This information is included on page 4 (lines 99-102) and new Fig. S1

3. The ventral turn on expression driven by LARM2 responding to ectopic expression of hLMX1B is a lovely confirmation that autoregulation, presumably via the identified sites, plays a part. However, the same experiment using the LARM1 construct is clearly missing.

We have now performed the equivalent experiment with *LARM1* that is included on page 5 (lines 109-111) and Fig. 2J. The experiment shows that *hLMX1B* is also sufficient to activate *LARM1* when the two are co-electroporated in the chicken ventral limb bud mesoderm.

4. Of the 5 SNVs identified in the nail-patella syndrome patient, it looks as if only one is actually within hLARM2- it seems important to show what effect just this one SNV has on expression.

We agree. The SNV within the *LARM2* region removes a predicted SP1 binding site. Site directed mutagenesis of just this sequence in the human construct to match the SNV, however, did not alter the *LARM2*'s activity in the chicken bioassay. We address this finding in the manuscript p 9 lines 207-209. The three SNVs 5' to *LARM2* substantially alter potential transcription factors binding sites. We suspect that disruption of these transcription factor binding can alter activity. However, the mechanism behind how the collective SNVs alter activity is unclear at present.

5. The chromatin marks in figure 1 suggest that Larm1 is the much more 'convincing' enhancer carrying (more) p300 and RNA pol2 but the human data implicates LARM2- Are both actually required? What happens to the limb phenotype when either is knocked out individually in the mouse?

The coexistence of positive and negative marks in *LARM2* likely reflects heterogeneity between dorsal and ventral cells that are mixed in the analyzed population (page 4, line 99). For the revised version of our manuscript we have generated the individual deletions of *LARM1* and *LARM2* (page 7 and new Fig. 6). For the individual deletion of *LARM2* we replicated the human deletion. Interestingly, we uncovered spatial modularity (*LARM1* predominant posterior influence, *LARM2* anterior influence) of these two enhancers. These results are included on pages 7 and 8 and in Fig. 6.

Minor points

Given that most of the data in figure 1 comes from published studies (identified in the M&Ms). it would be good to get matching references into the text.

Thank you for this comment, the appropriate references have been added whenever the published datasets are mentioned.

In figure 2, the brick red shading of the site in the delta4 construct is missing the final base. And in the figure legend, I think you've switched 3' and 5' in C).

Thank you very much for noting this, both mistakes have been corrected.

In Figure 3- would be worth pointing out these are all hindlimbs. Good to include some forelimb pictures too?

We emphasized in the figure legend that the images are of hindlimbs. We also include the forelimb phenotype as Fig. S3.

There's a discrepancy between the primer sequences in TableS1 and figure 2 – a few seem to have been switched and something's happened to the delta1 sequence (reversed?). Please fix.

Thank you. Yes, in the figure the sequence was actually 3' to 5' as noted above. The figure has been corrected and the sequence in the figure and the table appropriately match (now table S2).

Table S1- p2 – instead of primer sequences the text reads 'in the paper there are two sets of primers- were both used?' Please correct.

Thank you. This has been updated and Table S2-p2 now has the corrected primers that were used to sequence the *LARM* region.

In the M&M- in situ hybridisation reads 'To generate the chicken probe....' Please add the missing text.

This has been corrected, thank you.

Reviewer 2

The article entitled "Identification of limb-specific Lmx1b auto-regulatory modules with Nail-Patella Syndrome pathogenicity" characterizes two novel Lmx1b associated regulatory modules (LARM1 and LARM2) that likely encompass enhancers and silencers. Using epigenetic marks and ChIP-seq data, they identified several Lmx1b binding sites in candidate limb enhancers that are likely regulated Lmx1b expression. Indeed, they showed that these regulatory elements have similar spatiotemporal activity as the expression pattern of Lmx1b in the dorsal limb mesoderm. Using both chick and mouse enhancer assay, they have demonstrated the in vivo activity of each enhancer and the effect of mutated Lmx1b binding sites on the enhancer activity. Furthermore, while Lmx1b KO phenotype involved multiple tissues, the deletion of a region encompassing both LARM1 and LARM2 showed isolated limb phenotype, and also indicates the autoregulation role of Lmx1b on its own expression. Finally, they analyzed human NPS patients whom no coding mutations were found and test their regulatory sequences. They found two NPS families lacking coding sequence variations that had LARM2 variations that disrupt its activity and highlight the important role of regulatory modules in disease pathogenesis. In summary, this study showed Lmx1b regulatory elements that are auto-regulated by Lmx1b and this study also provides another evidence that noncoding SNP\deletion of transcriptional enhancers can alter gene regulation and lead to human diseases. I recommend publishing this study in NC after addressing the concerns below.

1. In the abstract, the authors describe functional KO of *Lmx1b* that cause decrease in its expression by nearly 6 fold. This sentence is confusing as the generated *Lmx1b* KO mouse was by a homozygous deletion of the gene, so how this gene can be transcribed?

If I understand what the authors wanted to point out is that the deletion of the *LARM1/2* reduces *Lmx1b* in the limb bud about 5-6 fold.

The *Lmx1b* KO mouse model was generated by deleting exons 3-7 of the *Lmx1b* gene (Chen et al., 1998, DOI: 10.1038/ng0598-51). In the null mutant, transcription of the truncated gene occurs allowing the quantification of transcription level. The truncated transcript is considered not to retain any significant *Lmx1b* function. We have added the *Lmx1b* levels detected in gene array data for e11.5 - e13.5 in table S1, which demonstrates the ~ 6 fold increased level of *Lmx1b* in normal mice at e12.5 (from Feenstra et al 2012).

2. The authors define that *LARM1s* is a silencer due to absent of GFP expression in 4 transgenic check embryos and by nucleotide substitution in the *Lmx1b* binding site that expanded the GFP expression to the entire limb. I think it is a good indication but additional functional assays are required. For example, to test if it is function as silencer with the *Lmx1b* promoter instead of the tk promoter and what should be the phenotype if only *LARM1s* will be deleted from the mouse genome.

The human *LMX1B* promoter is quite GC rich and our attempts to isolate it to further examine the silencer sequence in context of its own promoter were not successful. We did clone the silencer region into a reporter construct driven by the beta-Actin promoter. The beta-actin promoter is robust and the capacity of the *LARM1s* silencing region appears to be insufficient to noticeably reduce this promoter activity.

The generation of an animal model with the removal of just the silencer from *LARM1* is also an important question, but because of pandemic-related time constraints that have already extended us far beyond our expected resubmission time, we chose to focus on the individual removal of *LARM1* and *LARM2*, which were also requested.

Furthermore, the three nucleotide substitution in the *Lmx1b* (*LARM1-Δ1*) might generated a binding site for different transcription factor, so it not only eliminated *Lmx1b* consensus sequence but a different limb expressed TF activate the enhancer that lead to this observed GFP expression.

We generated two additional site directed mutagenesis of the *Lmx1b* binding site in the silencer region of *LARM1* to ensure that ventral extension is due to the disruption of the *Lmx1b* binding site rather than the addition of a new binding site (Fig S1). Please, see also response to Referee 1, points 1 and 2.

3. The authors have used the Capture-C data to show the interactions of *LARM1\2* region (Figure S1) but there is no clear explanation on tissue and developmental stage of this data. Moreover, it will be required to show the effect of the chromatin interactions in the homozygous *LARM* deletion versus WT mice.

The information on tissue and developmental stage (limb e11.5) has been included in the legend for Fig. S2. We considered performing 4C experiments, but this would require an enormous amount of cells and sequencing depth to reach the necessary resolution and confidence in interactions. Because of the considerable number of mice necessary to obtain the material and given that this is not on the core of our study, we have put these experiments on hold.

4. “Ectopic expression of human LMX1B in the ventral mesoderm is able to drive activity of the co-transfected LARM2-reporter construct (n=3)”. It is expected that LARM1 will be tested as well to show if it is also defended on the expression of LMX1B.

We ectopically expressed human LMX1B in the ventral mesoderm with the *LARM1*-reporter construct and demonstrated that LMX1B can drive *LARM1* activity (Fig 2J). Please, see also response to points 1 and 3 of Referee 1

5. In human, the deletion that cause the phenotype encompasses only LARM2 and not LARM1. Therefore, it is expected to model in mouse the deletion found in human by using the CRISPR system.

For the revised version of our manuscript we have generated the individual deletions of *LARM1* and *LARM2*. For the individual deletion of *LARM2* we replicated the human deletion. Interestingly, we uncovered spatial modularity (*LARM1* predominant posterior influence, *LARM2* anterior influence) of these two enhancers. These results are included on pages 7 and 8 and in Fig. 6.

6. The authors suggest that in the NPS case, the 5 SNPs are the cause for the phenotype. However, they did not show how well these SNPs are conserved. For instance, if these are conserved in mouse and check, it can indicate that these SNPs are more or less likely to be the cause. Furthermore, the nucleotide substitution does not overlap with the characterized LARM2 (except for one). This is very interesting results suggesting that the 5 SNPs can eliminate the activity of LARM2, when only one is overlapping. So, does this SNP it the functional one? Are the other 4 SNPs relevant for the LARM2 activity? If so, how they affect the LARM2 activity?

This is a great question. It was addressed, in part, in response to reviewer #1 (point 4), above. The SNV within the *LARM2* conserved region removes a predicted SP1 binding site. This site is highly conserved across most vertebrates including mice, but does not appear to be present in chicken as listed in the UCSC browser. As noted above, site directed mutagenesis of just this sequence in the human construct to match the SNV, did not alter the *LARM2*'s activity in the chicken bioassay. We address this finding in the manuscript p 9 lines 207-209. The SNV 3' to *LARM2* is also conserved, but not in chicken and does not appear to add or delete any transcription factor binding sites. The three SNVs 5' to *LARM2* are not well conserved, but all three substantially alter potential transcription factors binding sites. We suspect that disruption of transcription factor binding can both enhance and disrupt activity. However, the mechanism behind how these point mutations or SNVs alter enhancer activity is unclear at present and we agree requires further study. Although the characterization of this patient's *LARM2* region with 5 SNVs was not a primary focus of our paper, if the reviewer feels this information is important for evaluating the data will include it in the manuscript.

7. Lane 144:“Humans” should be without capital letter.

This has been corrected, thank you for pointing out.

Reviewer #3 (Remarks to the Author):

The manuscript by Haro et al. studies the cis-regulation of LMX1B, an essential gene for the proper establishment of the dorso-ventral axis in developing limbs. Building on previous research, the authors identify and dissect molecularly two upstream enhancers, termed LARM1 and LARM2, both directly bound by the Lmx1b protein. Using a chicken GFP-reporter system and through site-directed mutagenesis, the authors demonstrate the dorsally-restricted activity of the two regulatory elements, as well as the importance of Lmx1b-binding sites in driving enhancer function. Their findings are supported by the in vivo deletion of the enhancer region, causing a limb-restricted phenotype that recapitulates tissue-specific aspects of the full Lmx1b knockout. The study is nicely rounded with the identification of two unrelated human cases carrying mutations affecting the enhancer elements and displaying limb-restricted phenotypes.

This beautiful and important piece of work highlights the modular nature of regulatory elements in controlling tissue-specific aspects of gene expression and in building up complex developmental expression patterns. The manuscript is well written, clear, concise and with an appropriate interpretation and discussion of the data. For such reasons, I fully support its publication in Nature Communications.

I also provide suggestions for improvement that the authors might take into consideration:

Minor comments

1- In Figure S1, it is difficult to appreciate the overall TAD structure of the locus because of the low resolution of the Hi-C data as well as due to the region that was chosen to display.

I would recommend the inclusion of higher resolution data in the panel (Bonev et al 2017, Cell). In addition, the displayed region might be extended at least 1 Mb downstream (from 34,500,000 to 32,500,000) to fully appreciate the extension of the Lmx1b TAD.

Thank you. We have modified Fig. S1 following the referee suggestions. In the revised version of the manuscript former Fig. S1 is now Fig. S2.

2- The qPCR results on LARM KO mouse limbs shows residual expression of Lmx1b. This suggests that, although the Lmx1b enhancers are required to sustain limb expression levels, either the maintenance or initiation of expression is controlled by additional elements. Are there other elements near Lmx1b that could be performing such action? This aspect might be discussed briefly in the manuscript.

This is an excellent point. There does appear to be an additional potential *cis*-regulatory module (pCRM) about 10 kb further upstream from LARM2. Lmx1b does not appear to bind to this pCRM as it was not present in our Lmx1b ChIP-seq data, but it could be involved in the initial induction of *Lmx1b*, with LARM1/2 providing subsequent amplification to levels required for dorsalization. We have included this brief discussion on page 6 & 7 lines 150-154 and noted this pCRM in Fig 1A and Fig S2 (asterisk).

From Capture-C data, seems that Lmx1b interactions extends downstream of the gene. Figure 1 should be adapted accordingly to show the downstream interacting region and p300 and H3K27Ac ChIP-Seq tracks might be included in the upper panel, to display potential regulatory elements.

As highlighted by the reviewer, according to Capture-C experiments using the LMX1B promoter as a viewpoint where interactions seems to extend downstream of the gene, we agree that the addition of p300 and H3K27Ac in figure 1 would help to display additional regulatory regions in the LMX1b. However, we think that the addition of p300 and H3K27Ac will be more informative with the Capture-C data. Therefore, both tracks have been included in the in Fig S2 instead of Fig 1.

3- Lines 50-52. The assumption that haploinsufficiency accounts for the syndromic features is better sustained by the overlapping symptoms between missense, nonsense, or deletion mutations (Dunston et al., 2004, Genomics) or by in vitro experiments (Dreyer et al., 2000, Hum Mol Genet), rather than by interspecies differences as stated by the authors. I would reformulate this statement and cite these studies, to be more precise.

We absolutely agree. This sentence has been reformulated and references cited (page 3, lines 50-53, references 5 & 6)

4- Lines 183-185. This is a very interesting evolutionary hypothesis that might be worth discussing in moer detail. The LARM region is partially conserved in the fin-lobbed fish Coelacanth, a representative species of the transition from fins to limbs. Is this enhancer region also partially conserved in ray-finned fishes?

Figure 2 would benefit from the inclusion of genomic sequences of such fishes in the corresponding panels. In addition, it would be advisable to comment on the genomic differences (and the link with fin/limb morphology) between ray-finned fishes, lobe-finned fishes and tetrapods.

As noted, there is partial conservation in Coelacanth, and accordingly, we have included the genomic sequences of Coelacanth in Fig. 2B. However, there is no clear conservation in zebrafish

We agree with the referee in that this is a very attractive evolutionary hypothesis. The implication of DV limb polarity and the LARMs in the fin to limb transition is an ambitious currently ongoing project in the lab. Therefore, we decided to only very briefly comment on this issue in the discussion

5- It is unclear what marks the asterisk in Figure 1.

This is identifying the potential *cis*-regulatory module noted above in #2. We have referenced the asterisk in the legend for figures 1 and S2 and noted its potential as a *cis* regulatory module, p 14 lines 347-349.

6- In the legend of Figure S1, it is written “topographically” instead of “topologically”

Thank you, this has been corrected.

REVIEWERS' COMMENTS

Reviewer #1 (Remarks to the Author):

This was nice manuscript before revision- its a really lovely one now. The anterior/ posterior differences detected in the single LARM1/2 deletions are particularly interesting.

I'm happy to support publication.

Laura Lettice

Reviewer #2 (Remarks to the Author):

The revised article entitled "Identification of limb-specific Lmx1b auto-regulatory modules with Nail-Patella Syndrome pathogenicity" was dramatically improved and provide supporting evidence for their novel finding. They added new mouse models for both LARM1 and LARM2 that recapitulate a deletion that was found in a family with NPS. They also added transgenic mice emphasizing the activity of these LARM elements and suggested an evolutionary aspect for Lmx1b transcription regulation. However, few major points were not addressed in the revised manuscript. I recommend accepting this manuscript for publishing after addressing the following concerns:

1. The authors neglected a very important information about the regulation of Lmx1b locus. Taher et al. (PloS One, 2011), showed that C130021I20Rik is a novel gene co-transcribed from the Lmx1b promoter (Figure 3) with similar expression patten in the dorsal limb. It may reflect that the LARM1 and 2 regulate the transcription of this gene and not only Lmx1b. The authors should address this concern.
2. The authors have novel finding with a very interesting mechanism of enhancers and silencer\s that are regulated by Lmx1b to control gene expression. An example for such mechanism was recently published by Huang et al., 2021 (PMID: 33503407). In addition, the evolutionary of these sequences in fish is also interested. However, the abstract is missing this finding and it should be modified to be more informative and attractive about the findings.
3. "A possible interpretation is that the putative Lmx1b binding site within the 5' LARM1 element (TTATTA) can bind other transcription factors that silence the 3' LARM1 enhancer activity or

promote chromatin conformation that limits enhancer-promoter interaction." In recent years, many studies showed the important role of chromatin conformation in driving spatiotemporal expression, especially in development. Therefore, this is exactly the concern that should be address. The authors showed published capture C experiments showing that the LARM region physically interacts with the Lmx1b promoter (Fig. S2) but it was expected to perform 4C-seq or capture C on the limb of the mutants and compare them to WT. 4C-seq does not require an enormous amount of cells (as was done by others) and it can provide the necessary resolution and confidence for interactions.

4. "Interestingly, one of the enhancers of Apterous (ap), the Drosophila homologue of Lmx1b, is maintained by an autoregulatory loop, albeit indirectly through the ap targets vestigial and 129 scalloped (Vg/Sd) (Bieli et al, Plos Genetics).

Which enhancer? how this finding is relevant? the authors should clearly emphasis the support evidence and the meaning of the CRMs of the Lmx1b homologous in Drosophila. As in Bieli's paper (Figure 7G), the authors should suggest a mechanism that would explain how the identified elements play a role in chromatin organization at different stages of development in order to execute the expression required for limb development.

5. In Figure 6, the authors demonstrated the mild phenotype of the deletions of each LARM elements separately. It will be required to emphasis the extreme phenotype (mice can't walk) in LARM1 and 2 deletion was not found in these individual deletions. In addition to the in situ, it requires to show a reduction (or not) of Lmx1b expression in each model and compare it to the 6 fold reduction that was observed in null mice.

6. The enhancer assay in chick and mouse was done in several embryos (n=3-7, Figures 2 and 4). It is required to add in the supplementary the pictures of at least 3 embryos to evaluate the consistency of enhancer pattern and activity.

7. In the legend of figure 3, it is suggested to expand the description of pictures C-E as written in the results because it is not initiative for readers that are not expert in the anatomy of the limb.

Reviewer #3 (Remarks to the Author):

I would like to congratulate the authors for their efforts in the revision of the manuscript.

I have no further concerns.

Response to Reviewers

Reviewer 1

This was nice manuscript before revision- it's a really lovely one now. The anterior/ posterior differences detected in the single LARM1/2 deletions are particularly interesting.

I'm happy to support publication.

Thank you for your insight, suggestions, and support.

Reviewer 2

The authors neglected a very important information about the regulation of Lmx1b locus. Taher et al. (PloS One, 2011), showed that C130021I20Rik is a novel gene co-transcribed from the Lmx1b promoter (Figure 3) with similar expression pattern in the dorsal limb. It may reflect that the LARM1 and 2 regulate the transcription of this gene and not only Lmx1b. The authors should address this concern.

Thanks for bringing this to our attention. We have discussed this possible regulation on lines 129 -133 (page 6)

2. The authors have novel finding with a very interesting mechanism of enhancers and silencers that are regulated by Lmx1b to control gene expression. An example for such mechanism was recently published by Huang et al., 2021 (PMID: 33503407). In addition, the evolutionary of these sequences in fish is also interesting. However, the abstract is missing this finding and it should be modified to be more informative and attractive about the findings.

Thank you for alerting us to the paper by Huang et al., 2021. We have referenced this in our manuscript noting others have identified corepressors as a possible mechanism of silencing regional activity (see lines 108-110). We also thank the reviewer for the suggestion and we agree to add to the abstract the potential role of LARM enhancers and dorsalization in the limb-to-fin evolutionary transition (See lines 35-37).

3. "A possible interpretation is that the putative Lmx1b binding site within the 5' LARM1 element (TTATTA) can bind other transcription factors that silence the 3' LARM1 enhancer activity or promote chromatin conformation that limits enhancer-promoter interaction." In recent years, many studies showed the important role of chromatin conformation in driving spatiotemporal expression, especially in development. Therefore, this is exactly the concern that should be address. The authors showed published capture C experiments showing that the LARM region physically interacts with the Lmx1b promoter (Fig. S2) but it was expected to perform 4C-seq or capture C on the limb of the mutants and compare them to WT. 4C-seq does not require an enormous amount of cells (as was done by others) and it can provide the necessary resolution and confidence for interactions.

We thank the reviewer for the suggestion that additional chromatin conformation studies would provide insight into the spatiotemporal changes that may alter enhancer-promoter interactions. We agree these would be interesting, but at this time we are not set up to do these chromatin conformation experiments and others have already shown physical interactions between the LARM region and the

Lmx1b promoter, albeit without looking at potential spatiotemporal changes. We felt these experiments were outside the scope of our current study, but agreed that we wanted to demonstrate the potential for interaction. Thus, we used the published capture C experiments, to demonstrate the capacity for interaction, which was further supported by our knockout and in situ hybridization studies.

4. "Interestingly, one of the enhancers of Apterous (ap), the Drosophila homologue of Lmx1b, is maintained by an autoregulatory loop, albeit indirectly through the ap targets vestigial and scalloped (Vg/Sd) (Bieli et al, Plos Genetics).

Which enhancer? how this finding is relevant? the authors should clearly emphasize the support evidence and the meaning of the CRMs of the Lmx1b homologous in Drosophila. As in Bieli's paper (Figure 7G), the authors should suggest a mechanism that would explain how the identified elements play a role in chromatin organization at different stages of development in order to execute the expression required for limb development.

Thank you for your suggestion. We have clarified the enhancer (the apDV enhancer) and further clarified its relevance to our study (See lines 134-144).

5. In Figure 6, the authors demonstrated the mild phenotype of the deletions of each LARM elements separately. It will be required to emphasize the extreme phenotype (mice can't walk) in LARM1 and 2 deletion was not found in these individual deletions. In addition to the in situ, it requires to show a reduction (or not) of Lmx1b expression in each model and compare it to the 6 fold reduction that was observed in null mice.

This is a reasonable continuation of our discovery that LARM 1 and LARM2 regulate distal Lmx1b expression in a modular pattern (posterior and anterior, respectively). Additional studies comparing the combined LARM1/2 knockout with the individual LARM1 and LARM2 knockouts are warranted to characterize the module specific-activities of these two enhancers. Indeed, future investigations into these knockout models are being planned to determine the spatial and temporal activity of the two LARM enhancers. The quantitative analysis of Lmx1b expression suggested by the reviewer, although supportive, would not substantially change the story that the knockout phenotypes and the corresponding in situ hybridizations already demonstrate and we prefer to leave this for our in depth follow up study that will involve different stages and require careful dissections to separately analyze specific anterior-posterior regions.

6. The enhancer assay in chick and mouse was done in several embryos (n=3-7, Figures 2 and 4). It is required to add in the supplementary the pictures of at least 3 embryos to evaluate the consistency of enhancer pattern and activity.

We have compiled supplementary pictures of 3 embryos of each assay so consistency and the range of variation can be evaluated. This is supplemental figure S5a-d (chick) and S5e (mouse)

7. In the legend of figure 3, it is suggested to expand the description of pictures C-E as written in the results because it is not intuitive for readers that are not expert in the anatomy of the limb.

Thank you for your suggestion. We have expanded the figure legend for figure 3 and added additional lettering to further clarify the anatomy depicted (See lines 617-622).

Reviewer #3

I would like to congratulate the authors for their efforts in the revision of the manuscript.

I have no further concerns.

Thanks for your insight, suggestions and support.